# A Phase-Field Approach to Continuum Damage Mechanics

**DOI:** 10.3390/ma15217671

**Published:** 2022-10-31

**Authors:** Angelo Morro

**Affiliations:** DIBRIS, Università di Genova, 16145 Genova, Italy; angelo.morro@unige.it

**Keywords:** damage, aging, phase field, entropy production, 74D10, 74R05, 74C05, 74M25

## Abstract

This paper develops a phase-field approach to describe the damage within continuum mechanics. The body is associated with the standard stresses and body forces of macroscopic character. As is the case in many contexts, the phase field is a scalar variable whose time rate is governed by a constitutive equation. The generality of the approach allows the modeling of non-stationary heat conduction, mechanical hysteretic effects, and the macroscopic damage. The thermodynamic consistency is investigated through the constraint of the Clausius–Duhem inequality following the standard procedure of Rational Thermodynamics. The entropy production is considered as a constitutive function; this view was proved to be essential in the elaboration of hysteretic models. Here, the scheme is improved by viewing the entropy production as a sum of two terms, one induced by the other constitutive equations and one given by a constitutive equation of the entropy production per se.

## 1. Introduction

This paper is devoted to a thermodynamically consistent description of damage evolution in continuum mechanics. Damage in continuum mechanics is of interest in many respects. It is a way of characterizing the aging of materials, and it is also a measure used to describe the nucleation of new cracks in the absence of macroscopic initial cracks.

Perhaps motivated by the search of models more akin to the microscopic setting of damage, some approaches have been developed where micro stresses and micro forces are framed along with the standard stresses and forces of macroscopic character (see, e.g., [1,2,3,4,5]). Though this idea leads to a more detailed scheme, it involves additional unknown fields to be determined in an operative way. To make the approach simpler, micro kinetic terms are neglected and hence, micro forces are subject to equilibrium conditions. The purpose of this paper is to set up a phase-field approach to describe the damage within continuum mechanics. No microscopic fields are considered; the body is associated with the standard stresses and body forces of macroscopic character. As is the case in many contexts, the phase field (or order parameter or internal variable) is a scalar variable whose time rate is governed by a constitutive equation [6]. The generality of the present approach allows the modeling of non-stationary heat conduction, mechanical hysteretic effects, and macroscopic damage.

The thermodynamic consistency is investigated through the constraint of the Clausius–Duhem inequality following the standard procedure of Rational Thermodynamics [7]. Lately, a more general scheme has been applied in that the entropy production is considered as a constitutive function; this view has proved essential in the elaboration of hysteretic models [8,9,10]. Here, the scheme is improved by viewing the entropy production as a sum of two terms, one induced by the other constitutive equations and one given by a constitutive equation of the entropy production per se (extended entropy production). We will see that the occurrence of the extended entropy production is essential in connection with the damage variable with causes such as, e.g., excesses of temperature, strain, and stress.

Recent developments in material modeling show a clear distinction between different causes of damage. For instance, the displacement field is decomposed into translation, rotation, plastic stretches, elastic stretches, and volumetric and shear stretches [11,12]. The present approach is consistent with such more refined descriptions provided only that the entropy production, as any constitutive function, allows for the pertinent dependencies.

**Notation** **1.**We consider a solid occupying a time-dependent region $\Omega \subset E^{3}$. Throughout, v is the velocity, ∇ denotes the gradient operator, $\partial_{t}$ is the partial time derivative at a point x∈Ω, while a superposed dot stands for the total time derivative, $\dot{f}=\partial_{t}f+v\cdot \nabla f$. Cartesian coordinates are used: L is the velocity gradient, $L_{ij}=\partial_{x_{j}}v_{i}$, and D=symL is the stretching tensor. We let R be a reference configuration; the motion χ is a function that maps each point vector X∈R into a point x∈Ω. The deformation gradient F is defined by $F_{iK}=\partial_{X_{K}}\chi_{i}$ and J=detF>0. The other mathematical characters are defined at the first stage of usage.

## 2. Balance Equations and Admissible Processes

Let φ be the damage variable. According to the literature, there are several measures associated with the scalar φ. For instance, φ may be the fraction of damaged area [13] or the volumetric fraction of damaged material [5]. Anyway φ is a scalar with values in [0,1]; for definiteness, φ=0 represents an undamaged material, φ=1 a fully damaged material. There are cases in which the damage is anisotropic as it happens in elastostatics [12]. The scalar character of the damage variable might be maintained by considering a set of degradation functions, as seen in [14].

The body is modeled as a material with an internal variable (or phase field), the scalar φ. Hence, the balance equations are those of a simple body [15]. In the local form, the continuity equation, the balance of linear momentum, and the balance of energy are written as
(1)$$\dot{\rho }+\rho \nabla \cdot v=0,\;\rho \dot{v}=\nabla \cdot T+\rho b,\;\rho \dot{\varepsilon }=T\cdot D+\rho r-\nabla \cdot q,$$
where ρ is the mass density, v the velocity, T the symmetric stress tensor, b the body force (per unit mass), ε the internal energy density, q the heat flux, and *r* the external energy supply (per unit mass).

Let η be the entropy density (per unit mass). The balance of entropy is assumed in the form
(2)$$\rho \dot{\eta }+\nabla \cdot j-\frac{\rho r}{\theta }\geq 0,$$
where j is the entropy flux; we let j=q/θ+k with k the extra-entropy flux to be determined. In general, the entropy flux j need not equal q/θ, and is given by a constitutive equation. This view traces back to I. Müller [16] (see [17] for a detailed exposition).

We define the entropy production σ as
$$\sigma =\rho \dot{\eta }+\nabla \cdot j-\frac{\rho r}{\theta }$$
and, by the balance of entropy, σ≥0. Furthermore, we let σ be given by a constitutive equation, per se, as is the case for η and j. We let the set of fields ρ,v,T,b,ε,r,q and η,j (or k), and σ, subject to (Equation 1), be an *admissible process*. We take as the second law of thermodynamics the following statement.

**Second law of thermodynamics.** For every admissible process, the inequality (Equation 2) must hold for all times t∈R and points x∈Ω.

As is standard, we replace j with q/θ+k and substitute ρr−∇·q from the balance of energy (Equation 1)${}_{3}$ to rewrite inequality (Equation 2) in the form
$$-\rho \dot{\varepsilon }+\rho \theta \dot{\eta }+T\cdot D+\theta \nabla \cdot k-\frac{1}{\theta }q\cdot \nabla \theta =\theta \sigma \geq 0.$$

Using the Helmholtz free energy ψ=ε−θη, we obtain the inequality in the form
(3)$$-\rho \left(\dot{\psi }+\eta \dot{\theta }\right)+T\cdot D+\theta \nabla \cdot k-\frac{1}{\theta }q\cdot \nabla \theta =\theta \sigma \geq 0.$$

Additionally, for a connection with the literature, we mention other approaches to the modeling of damage.

### Other Balance Equations in the Literature

Motivated by a microscopic picture of damage, additional power terms are associated with φ via $\dot{\varphi }$ and $\nabla \dot{\varphi }$. By appealing to the principle of virtual power [18] the damage variable φ is taken to occur in the internal and external virtual powers $P_{i},P_{e}$. With reference to [19], we take
$$P_{i}\left(v,\dot{\varphi }\right)=-\int_{\Omega }T\cdot Ddv-\int_{\Omega }\left(\beta \dot{\varphi }+h\cdot \nabla \dot{\varphi }\right)dv$$
and
$$P_{e}\left(v,\dot{\varphi }\right)=\int_{\Omega }\rho b\cdot vdv+\int_{\partial \Omega }n\cdot Tvda+\int_{\Omega }\alpha \dot{\varphi }dv+\int_{\partial \Omega }\gamma \dot{\varphi }da.$$

Let $\varphi \to \varphi +\tilde{\varphi }$, $v\to v+\tilde{v}$. Hence, the requirement
$$P_{i}\left(\tilde{v},\tilde{\varphi }\right)+P_{e}\left(\tilde{v},\tilde{\varphi }\right)=0\;\forall \tilde{v},\forall \tilde{\varphi }$$
leads to
(4)α−β+∇·h=0,γ−h·n=0.

The microscopic fields α,β,h,γ are then subject to the local balance Equations (Equation 4). This in turn means that appropriate constitutive equations are needed.

Furthermore, the balance of energy is assumed to comprise the power $\beta \dot{\varphi }+h\cdot \nabla \dot{\varphi }$ so that the analogue of (Equation 1)${}_{3}$ would be [5]
$$\rho \dot{\varepsilon }=T\cdot D+\rho r-\nabla \cdot q+\beta \dot{\varphi }+h\cdot \nabla \dot{\varphi }.$$

The power $\beta \dot{\varphi }+h\cdot \nabla \dot{\varphi }$ would also occur in the entropy inequality [5].

We now go back to the present approach, which is free from fields of microscopic character.

## 3. Constitutive Equations and Thermodynamic Restrictions

The interest in constitutive dependencies on $\varphi ,\dot{\varphi }$, and $\nabla \dot{\varphi }$ indicates that the Lagrangian description is more convenient than the Eulerian one. Hence, we consider the referential quantities [15]
$$\rho_{R}=J\rho ,\;k_{R}=JkF^{-T},\;q_{R}=JqF^{-T},\;T_{RR}=JF^{-1}TF^{-T},$$
where $T_{RR}$ is the second Piola (or Piola–Kirchhoff) stress tensor, while
$$\;\nabla {\;}_{R}\;f\left(\chi \left(X\right)\right)=\nabla f\left(x\right)F,$$
for any function f(x). The multiplication of (Equation 3) by J=detF then results in the Lagrangian version of the second-law inequality,
(5)$$-\rho_{R}\left(\dot{\psi }+\eta \dot{\theta }\right)+T_{RR}\cdot \dot{E}-\frac{1}{\theta }q_{R}\cdot \nabla {\;}_{R}\;\theta +\theta \nabla {\;}_{R}\;\cdot k_{R}=J\theta \sigma ,$$
where $E=\frac{1}{2}\left(F^{T}F-1\right)$ is the Green–St. Venant strain tensor. The use of the referential quantities $T_{RR},E$, and $q_{R}$ is mathematically advantageous whenever we describe the material behavior through rate equations, in that ${\dot{T}}_{RR},\dot{E}$, and ${\dot{q}}_{R}$ are objective, whereas $\dot{T}$ and $\dot{q}$ are not. Of course (Equation 3) and (Equation 5) are equivalent. Though we use E, rather than $C=F^{T},F$, to describe the strain, the present approach applies to finite deformation in that no linear approximation is considered.

Let
$$\Gamma =(\theta ,E,T_{RR},q_{R},\varphi ,\dot{\theta },\dot{E},{\dot{T}}_{RR},{\dot{q}}_{R},\dot{\varphi },\nabla {\;}_{R}\;\theta ,\nabla {\;}_{R}\;\varphi )$$
be the set of independent variables. Compute $\dot{\psi }$ and substitute in (Equation 5) to obtain
$$\begin{array}{c} -\rho_{R}\left(\partial_{\theta }\psi +\eta \right)\dot{\theta }+\left(T_{RR}-\rho_{R}\partial_{E}\psi \right)\cdot \dot{E}-\rho_{R}\partial_{T_{RR}}\psi \cdot {\dot{T}}_{RR}-\rho_{R}\partial_{q_{R}}\psi \cdot {\dot{q}}_{R}-\rho_{R}\partial_{\varphi }\psi \dot{\varphi }\;\\ -\rho_{R}\partial_{\dot{\theta }}\psi \ddot{\theta }-\rho_{R}\partial_{\dot{E}}\psi \cdot \ddot{E}-\rho_{R}\partial_{{\dot{T}}_{RR}}\psi \cdot {\ddot{T}}_{RR}-\rho_{R}\partial_{{\dot{q}}_{R}}\psi \cdot {\ddot{q}}_{R}-\rho_{R}\partial_{\dot{\varphi }}\psi \ddot{\varphi }\;\\ -\rho_{R}\partial_{\nabla {\;}_{R}\;\theta }\psi \cdot \nabla {\;}_{R}\;\dot{\theta }-\rho_{R}\partial_{\nabla {\;}_{R}\;\varphi }\psi \cdot \nabla {\;}_{R}\;\dot{\varphi }-\frac{1}{\theta }q_{R}\cdot \nabla {\;}_{R}\;\theta +\theta \nabla {\;}_{R}\;\cdot k_{R}=J\theta \sigma \geq 0, \end{array}$$

The linearity and arbitrariness of $\ddot{E},{\ddot{T}}_{RR},{\ddot{q}}_{R},\ddot{\theta },\ddot{\varphi },\nabla {\;}_{R}\;\dot{\theta }$, and $\dot{\theta }$ imply that
$$\partial_{\dot{E}}\psi =0,\;\partial_{{\dot{T}}_{RR}}\psi =0,\;\partial_{{\dot{q}}_{R}}\psi =0,\;\partial_{\dot{\theta }}\psi =0,\;\partial_{\dot{\varphi }}\psi =0,\;\partial_{\nabla {\;}_{R}\;\theta }\psi =0,$$
and
(6)$$\eta =-\partial_{\theta }\psi .$$

Hence, the free energy is a function of a reduced number of variables, namely
$$\psi =\hat{\psi }\left(\theta ,E,T_{RR},q_{R},\varphi ,\nabla {\;}_{R}\;\varphi \right),$$
subject to the standard relation (Equation 6) for the entropy. For definiteness, we now examine further thermodynamic restrictions by specifying the type of continuum we have in mind.

Divide the remaining inequality by θ to get
(7)$$\begin{array}{c} \frac{1}{\theta }\left(T_{RR}-\rho_{R}\partial_{E}\psi \right)\cdot \dot{E}-\frac{\rho_{R}}{\theta }\partial_{T_{RR}}\psi \cdot {\dot{T}}_{RR}-\frac{\rho_{R}}{\theta }\partial_{\varphi }\psi \dot{\varphi }-\frac{\rho_{R}}{\theta }\partial_{\nabla {\;}_{R}\;\varphi }\psi \cdot \nabla {\;}_{R}\;\dot{\varphi }\\ -\frac{\rho_{R}}{\theta }\partial_{q_{R}}\psi \cdot {\dot{q}}_{R}-\frac{1}{\theta^{2}}q_{R}\cdot \nabla {\;}_{R}\;\theta +\nabla {\;}_{R}\;\cdot k_{R}=J\sigma \geq 0. \end{array}$$

In light of the identity
$$\frac{\rho_{R}}{\theta }\partial_{\nabla {\;}_{R}\;\varphi }\psi \cdot \nabla {\;}_{R}\;\dot{\varphi }=\nabla {\;}_{R}\;\cdot \left(\frac{\rho_{R}}{\theta }\partial_{\nabla {\;}_{R}\;\varphi }\psi \dot{\varphi }\right)-\nabla {\;}_{R}\;\cdot \left(\frac{\rho_{R}}{\theta }\partial_{\nabla {\;}_{R}\;\varphi }\psi \right)\dot{\varphi },$$
we can write inequality (Equation 7) in the form
$$\begin{array}{c} \frac{1}{\theta }\left(T_{RR}-\rho_{R}\partial_{E}\psi \right)\cdot \dot{E}-\frac{\rho_{R}}{\theta }\partial_{T_{RR}}\psi \cdot {\dot{T}}_{RR}-\frac{\rho_{R}}{\theta }\delta_{\varphi }\psi \;\dot{\varphi }-\frac{\rho_{R}}{\theta }\partial_{q_{R}}\psi \cdot {\dot{q}}_{R}\\ -\frac{1}{\theta^{2}}q_{R}\cdot \nabla {\;}_{R}\;\theta +\nabla {\;}_{R}\;\cdot \left(k_{R}-\frac{\rho_{R}}{\theta }\partial_{\nabla {\;}_{R}\;\varphi }\psi \;\dot{\varphi }\right)=J\sigma \geq 0, \end{array}$$
where
$$\delta_{\varphi }\psi =\partial_{\varphi }\psi -\frac{\theta }{\rho_{R}}\nabla {\;}_{R}\;\cdot \left(\frac{\rho_{R}}{\theta }\partial_{\nabla {\;}_{R}\;\varphi }\psi \right)$$
is the variational derivative of ψ with respect to φ. This suggests that we let
$$k_{R}=\frac{\rho_{R}}{\theta }\partial_{\nabla {\;}_{R}\;\varphi }\psi \;\dot{\varphi },$$
where $\dot{\varphi }$ is yet to be determined.

For definiteness, we take $\nabla {\;}_{R}\;\theta$ and ${\dot{q}}_{R}$ as independent of $\dot{E},{\dot{T}}_{RR},\dot{\varphi }$ and let
$$\sigma_{\mathrm{mech}}{=\sigma |}_{{\dot{q}}_{R}=0,\nabla {\;}_{R}\;\theta =0},$$
$$\sigma_{q}{=\sigma |}_{\dot{E}=0,{\dot{T}}_{RR}=0,\dot{\varphi }=0},$$
where ${\sigma |}_{\cdot }$ denotes the appropriate restriction of the function σ. Hence, we can split the inequality to get
(8)$$\left(T_{RR}-\rho_{R}\partial_{E}\psi \right)\cdot \dot{E}-\rho_{R}\partial_{T_{RR}}\psi \cdot {\dot{T}}_{RR}-\rho_{R}\delta_{\varphi }\psi \;\dot{\varphi }=J\theta \sigma_{\mathrm{mech}}\geq 0,$$
(9)$$-\rho_{R}\partial_{q_{R}}\psi \cdot {\dot{q}}_{R}-\frac{1}{\theta }q_{R}\cdot \nabla {\;}_{R}\;\theta =J\theta \sigma_{q}\geq 0.$$

### 3.1. Further Thermodynamic Restrictions

The damaged material has properties affected by the level of damage, formally characterized by φ. We then look for a modeling with the free energy in the form
$$\psi =\psi_{\mathrm{el}}\left(\theta ,\varphi ,E,T_{RR}\right)+\psi_{\mathrm{dam}}\left(\theta ,\varphi ,\nabla {\;}_{R}\;\varphi \right)+\psi_{q}\left(\theta ,\varphi ,q_{R}\right).$$

We begin with Equation (Equation 9), which governs heat conduction.

#### 3.1.1. Heat Conduction

The function $\psi_{q}$ is a scalar function of $q_{R}$. We then let $\psi_{q}$ depend on $q_{R}$ through $\xi =|q_{R}{|}^{2}$. Hence, Equation (Equation 9) can be written in the form
(10)$$-\left(2\rho_{R}\partial_{\xi }\psi \;{\dot{q}}_{R}+\frac{1}{\theta }\nabla {\;}_{R}\;\theta \right)\cdot q_{R}=J\theta \sigma_{q}\geq 0,$$
where $\sigma_{q}$ is independent of $\dot{E},{\dot{T}}_{RR},\dot{\varphi }$.

The requirement (Equation 10) implies that $q_{R},{\dot{q}}_{R}$, and $\nabla {\;}_{R}\;\theta$ cannot be independent. Among the relations consistent with (Equation 10), we select
(11)$$2\rho_{R}\partial_{\xi }\psi_{q}\;{\dot{q}}_{R}+\frac{1}{\theta }\nabla {\;}_{R}\;\theta =-\frac{1}{\kappa \theta }q_{R},\;2\rho_{R}\partial_{\xi }\psi_{q}\;{\dot{q}}_{R}+\frac{1}{\theta }\nabla {\;}_{R}\;\theta =-\frac{1}{\kappa \theta }Cq_{R}$$
so that in stationary conditions, we have Fourier-like laws
$$q_{R}=-\kappa \nabla {\;}_{R}\;\theta ,\;q_{R}=-\kappa C^{-1}\nabla {\;}_{R}\;\theta .$$

We then obtain, respectively,
$$\sigma_{q}=\frac{1}{\kappa J\theta^{2}}{\left|q_{R}\right|}^{2},\;\sigma_{q}=\frac{1}{\kappa J\theta^{2}}q_{R}\cdot Cq_{R},$$
where κ is allowed to depend on θ and φ. The non-negative character of $\sigma_{q}$ implies that κ>0.

To obtain the corresponding Eulerian version, we observe that
$$\nabla {\;}_{R}\;\theta =F^{T}\nabla \theta ,\;q_{R}=JF^{-1}q,\;{\dot{q}}_{R}=JF^{-1}\overset{☐}{q},$$
where $\overset{☐}{q}$ is the Truesdell derivative [20,21]
$$\overset{☐}{q}=\dot{q}-Wq-Dq+\left(\nabla \cdot v\right)q.$$

Hence, it follows from (Equation 11)${}_{1}$ that
(12)$$2\theta \rho_{R}\kappa \partial_{\xi }\psi_{q}\overset{☐}{q}+q=-\left(\kappa /J\right)B\nabla \theta .$$

As to (Equation 11)${}_{2}$ we have
(13)$$CF^{-1}q+2\theta \rho_{R}\kappa \partial_{\xi }\psi_{q}F^{-1}\overset{☐}{q}=-\left(\kappa /J\right)F^{T}\nabla \theta .$$

Left multiplication of (Equation 13) by $FC^{-1}$ results in
(14)$$q+2\theta \rho_{R}\kappa \partial_{\xi }\psi_{q}B^{-1}\overset{☐}{q}=-\left(\kappa /J\right)\nabla \theta .$$

In both cases, we can view
$$\tau =2\theta \rho_{R}\kappa \partial_{\xi }\psi_{q}$$
as the relaxation time. Both Equations (Equation 11) can be viewed as nonlinear generalizations of the Maxwell–Cattaneo equation [22,23]. The occurrence of the left Cauchy–Green tensor B in (Equation 12) and (Equation 14) shows possible effects of deformation on the heat conduction. In both cases, in stationary conditions ($\overset{☐}{q}=0$), we have the classical Fourier law, $q=-\tilde{\kappa }\nabla \theta$, or the modification due to deformation, $q=-\tilde{\kappa }B\nabla \theta$, where $\tilde{\kappa }=\kappa /J$.

There are infinitely many free energies $\psi_{q}$ consistent with this scheme. If, for simplicity, we let
$$\psi_{q}=\frac{1}{2}\nu \left(\theta \right){\left|q_{R}\right|}^{2}$$
then
$$\tau =\theta \rho_{R}\kappa \nu .$$

We now go back to Equation (Equation 8) and assume $\dot{\varphi }$ is independent of $\dot{E}$ and ${\dot{T}}_{RR}$ so that we can write the independent equations
(15)$$\rho \delta_{\varphi }\psi \;\dot{\varphi }=-\theta \sigma_{\mathrm{dam}}\leq 0,$$
(16)$$\left(T_{RR}-\rho_{R}\partial_{E}\psi_{\mathrm{el}}\right)\cdot \dot{E}-\rho_{R}\partial_{T_{RR}}\psi_{\mathrm{el}}\cdot {\dot{T}}_{RR}=J\theta \sigma_{\mathrm{visco}}\geq 0.$$

#### 3.1.2. Hysteretic Mechanical Effects

Some classes of materials models described by (Equation 16) are now investigated. First, we consider the particular case $\sigma_{\mathrm{visco}}=0$ and $\partial_{T_{RR}}\psi_{\mathrm{el}}=0$. The resulting equation is
$$\left(T_{RR}-\rho_{R}\partial_{E}\psi_{\mathrm{el}}\right)\cdot \dot{E}=0.$$

The arbitrariness of $\dot{E}$ implies that
$$T_{RR}=\rho_{R}\partial_{E}\psi_{\mathrm{el}}\left(\theta ,\varphi ,E\right).$$

This relation models an elastic solid parameterized by the temperature θ and the damage variable φ.

If instead $\sigma_{\mathrm{visco}}=0$, but $\partial_{T_{RR}}\psi_{\mathrm{el}}\neq 0$, then we have
$$\left(T_{RR}-\rho_{R}\partial_{E}\psi_{\mathrm{el}}\right)\cdot \dot{E}-\rho_{R}\partial_{T_{RR}}\psi_{\mathrm{el}}\cdot {\dot{T}}_{RR}=0.$$

Consequently, ${\dot{T}}_{RR}$ can be given a linear representation in $\dot{E}$. Indeed, we use a representation formula of tensors [10]; for any tensor Z with a known value of the inner product Z·N, and N·n=1, we have
(17)Z=(Z·N)N+(I−N⊗N)G,
where I is the fourth-order identity and G is an arbitrary second-order tensor. Here, we let $Z={\dot{T}}_{RR}$ and $N=\rho_{R}\partial_{T_{RR}}\psi_{\mathrm{el}}/\left|\rho_{R}\partial_{T_{RR}}\psi_{\mathrm{el}}\right|$ to obtain
$${\dot{T}}_{RR}=\left(\frac{1}{\rho_{R}}T_{RR}-\partial_{E}\psi_{\mathrm{el}}\right)\cdot \dot{E}\;\frac{\partial_{T_{RR}}\psi_{\mathrm{el}}}{|\partial_{T_{RR}}\psi_{\mathrm{el}}{|}^{2}}+\left(I-\frac{\partial_{T_{RR}}\psi_{\mathrm{el}}⊗\partial_{T_{RR}}\psi_{\mathrm{el}}}{|\partial_{T_{RR}}\psi_{\mathrm{el}}{|}^{2}}\right)G.$$

Observe that by letting $G=G_{RR}\left(\theta ,\varphi ,E,T_{RR}\right)\dot{E}$, we can write
(18)$${\dot{T}}_{RR}=C_{RR}\dot{E},$$
where
$$C_{RR}=G_{RR}+\frac{1}{\rho_{R}{\left|\partial_{T_{RR}}\psi \right|}^{2}}\partial_{T_{RR}}\psi ⊗\left(T_{RR}-\rho_{R}\partial_{E}\psi -\rho_{R}G_{RR}^{T}\partial_{T_{RR}}\psi \right).$$

The representation (Equation 18) describes the constitutive properties of *hypoelastic* solids.

Elastic–plastic models are characterized by an entropy production—here, $\sigma_{\mathrm{visco}}$—which depends linearly on $|\dot{E}|$ or $|{\dot{T}}_{RR}|$. Back to (Equation 16); let $J\theta \sigma_{\mathrm{visco}}=\gamma_{E}\left|\dot{E}\right|$ to obtain
$$\left(T_{RR}-\rho_{R}\partial_{E}\psi_{\mathrm{el}}\right)\cdot \dot{E}-\rho_{R}\partial_{T_{RR}}\psi_{\mathrm{el}}\cdot {\dot{T}}_{RR}=\gamma_{E}\left|\dot{E}\right|.$$

Assume $\partial_{T_{RR}}\psi_{\mathrm{el}}\neq 0$. Hence, we can write
$${\dot{T}}_{RR}\cdot \frac{\partial_{T_{RR}}\psi_{\mathrm{el}}}{\rho_{R}\left|\partial_{T_{RR}}\psi_{\mathrm{el}}\right|}=\frac{1}{\rho_{R}\left|\partial_{T_{RR}}\psi_{\mathrm{el}}\right|}\left(T_{RR}-\rho_{R}\partial_{E}\psi_{\mathrm{el}}\right)\cdot \dot{E}-\frac{\gamma_{E}}{\rho_{R}\left|\partial_{T_{RR}}\psi_{\mathrm{el}}\right|}\left|\dot{E}\right|.$$

Letting
$$P_{RR}=\frac{\partial_{T_{RR}}\psi }{\rho_{R}{\left|\partial_{T_{RR}}\psi \right|}^{2}}$$
and applying the representation formula (Equation 17), we obtain
(19)$${\dot{T}}_{RR}=C_{RR}\dot{E}-\gamma_{E}P_{RR}\left|\dot{E}\right|.$$

The analogue holds if we let $\sigma_{\mathrm{visco}}=\gamma_{T}\left|{\dot{T}}_{RR}\right|$.

As is shown in refs [8,9,10], in one-dimensional settings, the simultaneous occurrence of $\dot{E}$ and $|\dot{E}|$ allows the modeling of hysteretic processes. Here, we have proved that the structure (Equation 19) follows directly from the entropy inequality by simply letting the positive quantity σ equal $\gamma_{E}\left|\dot{E}\right|$ or $\gamma_{T}\left|{\dot{T}}_{RR}\right|$. Moreover, this value of σ is not identically equal to the left-hand side of the entropy inequality, as instead it happens for the model of heat conduction, where $\sigma_{q}$ is determined by the left-hand side of (Equation 11). This conceptual aspect will be more refined, in the next section, in connection with the modeling of damage.

#### 3.1.3. Damage and Degradation

We now investigate (Equation 15), which is the main equation describing the evolution of the damage. Let
$$\psi =\Psi :=G\left(\varphi \right)\tilde{\psi }\left(\theta ,E,T_{RR},q_{R}\right)+\psi_{\mathrm{dam}}\left(\theta ,\varphi ,\nabla {\;}_{R}\;\varphi \right).$$

Hence, Equation (Equation 15) results in
(20)$$\rho \left(G^{\prime }\left(\varphi \right)\tilde{\psi }+\delta_{\varphi }\psi_{\mathrm{dam}}\right)\dot{\varphi }=-\theta \sigma_{\mathrm{dam}}\leq 0.$$

The function G(φ) models the degradation induced by damage. Since φ=0 characterizes the undamaged state, the function *G* is subject to G(0)=1,G(1)=0. In addition, observe that
$$G^{\prime }\left(\varphi \right)\tilde{\psi }\;\dot{\varphi }=\dot{G}\left(\varphi \right)\tilde{\psi }.$$

We expect that the thermoelastic properties decay with the increase in the damage variable φ, and hence
$$\dot{G}\leq 0,\;\dot{\varphi }\geq 0\;⟹\;G^{\prime }\leq 0.$$

This qualitative conclusion is consistent with the observation that, by (Equation 20), $\mathrm{sgn}\left\{G^{\prime }\dot{\varphi }\right\}=\mathrm{sgn}\left\{-\sigma_{\mathrm{dam}}\right\}$, not necessarily but consistently. With this in mind, we can view *G* as a known function subject to the monotone character $G^{\prime }\leq 0$. Our attention is then restricted to $\psi_{\mathrm{dam}}$, a function of the values φ and $\nabla {\;}_{R}\;\varphi$; the dependence on $\nabla {\;}_{R}\;\varphi$ represents the possible effects of spatial inhomogeneities.

We now consider Equation (Equation 15) in the form
(21)$$\delta_{\varphi }\Psi \dot{\varphi }=-\frac{\theta }{\rho }\sigma_{\mathrm{dam}}\leq 0.$$

The left-hand side is non-negative if
(22)$$\dot{\varphi }=-\lambda \left(\theta ,\varphi \right)\delta_{\varphi }\Psi ,\;\lambda \geq 0.$$

Equation (Equation 21) allows for a further contribution to $\dot{\varphi }$ so that
(23)$$\dot{\varphi }=-\lambda \delta_{\varphi }\Psi -\frac{\theta }{\rho \delta_{\varphi }\Psi }{\hat{\sigma }}_{\mathrm{dam}}.$$

This is so that
$$\rho \delta_{\varphi }\Psi \dot{\varphi }=-\lambda \rho {\left(\delta_{\varphi }\Psi \right)}^{2}-\theta {\hat{\sigma }}_{\mathrm{dam}},$$
which shows that the rate Equation (Equation 23) yields an entropy production
$$\sigma_{\mathrm{dam}}={\hat{\sigma }}_{\mathrm{dam}}+\lambda \frac{\rho }{\theta }{\left(\delta_{\varphi }\Psi \right)}^{2}.$$

Look at the two effects separately. First, let ${\hat{\sigma }}_{\mathrm{dam}}=0$. Additionally, with reference to the literature (see e.g., [4,5] and refs therein) we may take λ as a constant, possibly related to parameters of the material (here, $\theta ,E,T_{RR},q_{R}$), and
$$\psi_{\mathrm{dam}}=\psi_{1}\left(\varphi \right)+\psi_{2}(|\nabla {\;}_{R}\;\varphi {|}^{2}).$$

Hence, the evolution Equation (Equation 22) reads
$$\dot{\varphi }=-\lambda \{\psi_{1}^{\prime }\left(\varphi \right)-2\frac{\theta }{\rho }\nabla {\;}_{R}\;\cdot [\frac{\rho }{\theta }\psi_{2}^{\prime }(|\nabla {\;}_{R}{\;\varphi |}^{2})\nabla {\;}_{R}\;\varphi ]\}-\lambda G^{\prime }\tilde{\psi }.$$

If, in particular, $\psi_{2}(|\nabla {\;}_{R}{\;\varphi |}^{2})=\frac{1}{2}c{\left|\nabla {\;}_{R}\;\varphi \right|}^{2}$ and ρ,θ are space independent, then we have
$$\dot{\varphi }=-\lambda \left\{\psi_{1}^{\prime }\left(\varphi \right)-c\Delta_{R}\varphi \right\}-\lambda G^{\prime }\tilde{\psi }.$$

Incidentally, in these cases,
$$\theta \sigma_{\mathrm{dam}}=-\delta_{\varphi }\psi_{\mathrm{dam}}\;\dot{\varphi }=\lambda {\left(\delta_{\varphi }\psi_{\mathrm{dam}}\right)}^{2}$$
and hence the requirement $\sigma_{\mathrm{dam}}\geq 0$ holds for every function $\psi_{\mathrm{dam}}$.

We now look for specific forms of ${\hat{\sigma }}_{\mathrm{dam}}$. Suppose that an increase of φ is due to high temperatures, $\theta >\theta_{h}$, freeze–thaw transitions at $\theta =\theta_{0}$, and large strains |E|>κ. Large cycles are also of interest. Letting E=Ee⊗e, we have damage effects if $E>E_{+}>0$ or $E<E_{-}<0$ for suitable values $E_{-},E_{+}$. Effects of large values are modeled through terms proportional to
$$H\left(\theta -\theta_{h}\right),\;H\left(|E|-\kappa \right),\;H\left(E-E_{+}\right),\;H\left(E_{-}-E\right),$$

*H* being the Heaviside step function. The whole effect on ${\hat{\sigma }}_{\mathrm{dam}}$ can then be written via appropriate coefficients in the form
(24)$${\hat{\sigma }}_{\mathrm{dam}}=c_{h}H\left(\theta -\theta_{h}\right)+c_{E}H\left(|E|-\kappa \right)+c_{+}H\left(E-E_{+}\right)+c_{-}H\left(E_{-}-E\right).$$

## 4. History Effects on Damage

Damage may be the effect of phase transformations, as, e.g., in the solid–liquid case, or cyclic processes, as e.g., in periodic or hysteretic evolutions. This indicates that the present value of φ depends on the history of appropriate fields. The idea is not new (see, e.g., [5]). The subject deserves a detailed treatment both for conceptual questions associated with the thermodynamic consistency and for the development of definite models.

We follow a Lagrangian description and suppose the constitutive properties are expressed by the set of variables
$$\Gamma =(\theta ,\varphi ,\theta^{t},\nabla {\;}_{R}\;\theta ,q_{R},E,T_{RR},E^{t}),$$
where $\theta^{t},E^{t}$ denote the histories of θ,E up to time *t*,
$$\theta^{t}\left(s\right)=\theta \left(t-s\right),\;E^{t}\left(s\right)=E\left(t-s\right),\;s\in \left[0,\infty \right).$$

We then let ψ,η,k,σ, and the rate $\dot{\varphi }$ be functionals of Γ. Upon computing $\dot{\psi }$ we replace in the Clausius–Duhem inequality (Equation 5) to obtain
$$\begin{array}{c} -\rho_{R}\left(\partial_{\theta }\psi +\eta \right)\dot{\theta }+\left(T_{RR}-\rho_{R}\partial_{E}\psi \right)\cdot \dot{E}-\rho_{R}\partial_{T_{RR}}\psi \cdot {\dot{T}}_{RR}-\rho_{R}\partial_{q_{R}}\psi \cdot {\dot{q}}_{R}-\rho_{R}\partial_{\varphi }\psi \dot{\varphi }\;\\ -\rho_{R}\partial_{\nabla {\;}_{R}\;\theta }\psi \cdot \nabla {\;}_{R}\;\dot{\theta }-\rho d\psi \left(\theta^{t}|{\dot{\theta }}^{t}\right)-\rho d\psi \left(E^{t}|{\dot{E}}^{t}\right)-\frac{1}{\theta }q_{R}\cdot \nabla {\;}_{R}\;\theta +\theta \nabla {\;}_{R}\;\cdot k_{R}=J\theta \sigma \geq 0, \end{array}$$
where $d\psi (\theta^{t}|{\dot{\theta }}^{t})$ and $d\psi (E^{t}|{\dot{E}}^{t})$ denote the Fréchet derivatives of ψ at $\theta^{t}$ and $E^{t}$ in the direction ${\dot{\theta }}^{t}$ and ${\dot{E}}^{t}$.

The reduced inequality is
(25)$$\begin{array}{c} \left(T_{RR}-\rho_{R}\partial_{E}\psi \right)\cdot \dot{E}-\rho_{R}\partial_{T_{RR}}\psi \cdot {\dot{T}}_{RR}-\rho_{R}\partial_{q_{R}}\psi \cdot {\dot{q}}_{R}-\frac{1}{\theta }q_{R}\cdot \nabla {\;}_{R}\;\theta -\rho_{R}\partial_{\varphi }\psi \dot{\varphi }\;\\ -\rho d\psi \left(\theta^{t}|{\dot{\theta }}^{t}\right)-\rho d\psi \left(E^{t}|{\dot{E}}^{t}\right)+\theta \nabla {\;}_{R}\;\cdot k_{R}=J\theta \sigma \geq 0. \end{array}$$

Observe that
$$\left(T_{RR}-\rho_{R}\partial_{E}\psi \right)\cdot \dot{E}-\rho_{R}\partial_{T_{RR}}\psi \cdot {\dot{T}}_{RR}$$
allows us to describe hysteretic effects,
$$-\rho_{R}\partial_{q_{R}}\psi \cdot {\dot{q}}_{R}-\frac{1}{\theta }q_{R}\cdot \nabla {\;}_{R}\;\theta$$
models heat conduction, and
$$-\rho_{R}\partial_{\varphi }\psi \dot{\varphi }$$
indicates the evolution of damage. A sufficient condition to satisfy (Equation 25) is to let
$$k_{R}=0,\;\psi \;\mathrm{independent}\;\mathrm{of}\;\theta^{t},E^{t}$$
and
$$\sigma =\sigma_{\mathrm{hys}}\left(\dot{E},{\dot{T}}_{RR}\right)+\sigma_{\mathrm{cond}}\left(q_{R}\right)+\sigma_{\mathrm{dam}}\left(\theta^{t},E^{t}\right)$$
$\sigma_{\mathrm{hys}},\;\sigma_{\mathrm{cond}},\;\sigma_{\mathrm{dam}}$ being non-negative in that they are particular cases of σ. Hence, in addition to the equations for $\sigma_{\mathrm{hys}},\sigma_{\mathrm{cond}}$, we have
$$\rho_{R}\partial_{\varphi }\psi \dot{\varphi }=J\theta \sigma_{\mathrm{dam}}\left(\theta^{t},E^{t}\right).$$

For definiteness, we look for specific forms of $\sigma_{\mathrm{dam}}$. As observed above, large values effects are modeled through terms proportional to
$$H\left(\theta -\theta_{h}\right),\;H\left(|E|\right)>\kappa ,\;H\left(E-E_{+}\right),\;H\left(E_{-}-E\right).$$

A model for the effect of a freeze–thaw transition is made formal by letting δ>0 and taking a contribution, at time *t*, of the form
$$H[\left(\theta \left(t-\delta \right)-\theta_{0}\right)\left(\theta_{0}-\theta \left(t\right)\right)]$$
which works for freeze–thaw and for thaw–freeze. The whole change in φ as $t\in [t_{1},t_{2}]$ is then expressed by
(26)$$\begin{array}{c} \varphi \left(t_{2}\right)-\varphi \left(t_{1}\right)=\int_{t_{1}}^{t_{2}}\frac{\theta }{\rho \partial_{\varphi }\psi }[c_{h}H\left(\theta -\theta_{h}\right)+c_{E}H\left(|E|-\kappa \right)\;\\ +c_{ft}\frac{1}{\delta }H\left[\left(\theta \left(t-\delta \right)-\theta_{0}\right)\left(\theta_{0}-\theta \left(t\right)\right)\right]+c_{+}H\left(E-E_{+}\right)+c_{-}H\left(E_{-}-E\right)]dt. \end{array}$$

A model for the effect of a freeze–thaw transition is made formal by letting δ>0 and taking a contribution, at time *t*, of the form
$$H[\left(\theta \left(t-\delta \right)-\theta_{0}\right)\left(\theta_{0}-\theta \left(t\right)\right)]$$
which works for freeze–thaw and for thaw–freeze.

For the sake of simplicity, sometimes the damage variable is evaluated by a cumulative assessment (Miner’s rule [24,25]). So, if the damage is produced by a number of stress cycles, at a given stress level, and *N* is the number of cycles producing failure, then φ associated with n≤N cycles is determined by
$$\varphi =\frac{n}{N}.$$

Incidentally, for each freeze–thaw transition at time $\tilde{t}$, we have
$$\frac{1}{\delta }\int_{\tilde{t}}^{\tilde{t}+\delta }H\left[\left(\theta \left(t-\delta \right)-\theta_{0}\right)\left(\theta_{0}-\theta \left(t\right)\right)\right]dt=1$$
and then
$$\frac{1}{\delta }\int_{t_{1}}^{t_{2}}H\left[\left(\theta \left(t-\delta \right)-\theta_{0}\right)\left(\theta_{0}-\theta \left(t\right)\right)\right]dt=n,$$
the number of freeze–thaw transitions as $t\in [t_{1},t_{2}]$.

Once we fix the derivative $\partial_{\varphi }\psi$ and the coefficients $c_{h},c_{E},c_{ft},c_{+},c_{-}$, we obtain the variation of the damage variable φ in $\left[t_{1},t_{2}\right]$.

## 5. Remarks on the Phase-Field Models

The evolution Equation (Equation 24) generalizes known model equations appeared in the literature. The rate $\dot{\varphi }$ consists of two parts,
$$-\lambda \delta_{\varphi }\Psi ,\;-\frac{\theta }{\rho \delta_{\varphi }\Psi }{\hat{\sigma }}_{\mathrm{dam}}.$$

The first term involves the present value of φ, possibly through the Laplacian $\Delta_{R}\varphi$. With reference to the review paper [26], as particular cases we mention the models by Karma et al., Henry and Levine, Ambati et al., and Miehe et al., where $\dot{\varphi }$ is affected by φ,Δφ, and is governed by the strain.

The second term allows specific causes of damage. Equation (Equation 24), and the analogue (Equation 26) for modeling through histories, indicates how the model equation of $\dot{\varphi }$ may account for high temperatures, freeze–thaw transitions, large strains, and cycles. The possible dependence of the coefficients $c_{h},c_{E},c_{+},c_{-}$ on φ itself and the temperature θ allows an effective modeling of damage effects. Experimental evidence might give insights into the constitutive dependencies of the coefficients.

### Relation to Other Approaches

A number of papers involve the use of microforces in the balance equations. Nevertheless, some similarities appear. In [3], the free energy is assumed in the form
$$\psi \left(E,\varphi ,\nabla \varphi \right)=\frac{1}{2}G\left(\varphi \right)E\cdot CE+G(\frac{w(\varphi )}{4\epsilon }+{\epsilon |\nabla \varphi |}^{2}),$$
which shows the correspondence
$$\tilde{\psi }=\frac{1}{2}E\cdot CE,\;\psi_{\mathrm{dam}}=G(\frac{w(\varphi )}{4\epsilon }+{\epsilon |\nabla \varphi |}^{2}),$$
G and ϵ as parameters. The function
$$w\left(\varphi \right)=16\varphi^{2}{\left(1-\varphi \right)}^{2}$$
describes the double well potential. This potential is widely used in the modeling of phase transitions. To our mind, in the modeling of damage, a monotone dependence would be more appropriate; the monotone character is advocated, e.g., in [27].

As in [5], the free energy is sometimes taken to depend on the history of $E^{t}$ and, moreover, the free energy potential is considered in terms of a fractional derivative. The thermodynamic consistency requires that the derivative $d\psi (E^{t}|{\dot{E}}^{t})$ is among the contributions to the non-negative entropy production. Now, while a free energy potential can be written for the stress tensor, problems arise when we investigate the thermodynamic consistency, mainly because the kernel of fractional derivatives is singular at the origin [28].

## 6. Conclusions

A model for the characterization of damage in continuum mechanics is developed for a dissipative and heat-conducting solid. The damage is described by a scalar variable φ (phase field) whose evolution is governed by a rate equation consistent with thermodynamics. The consistency is investigated within a differential equation of the form (Equation 23),
$$\dot{\varphi }=-\lambda \delta_{\varphi }\Psi -\frac{\theta }{\rho \delta_{\varphi }\Psi }\hat{\sigma },\;\lambda >0,$$
where Ψ is the free energy density. The term $\lambda \delta_{\varphi }\Psi$ is similar to models in the literature. The second term allows an account of effects such as those arising from large temperatures, large strains, and freeze–thaw transitions.

Conceptually, the two terms have a different origin. From
$$\delta_{\varphi }\Psi \dot{\varphi }\leq 0,$$
we conclude that $\dot{\varphi }$ may be given by $-\lambda \delta_{\varphi }\Psi$, λ>0. This in turn implies that the corresponding entropy production is ρ/θ times $\lambda {\left(\delta_{\varphi }\Psi \right)}^{2}$. This value of entropy production is induced by the constitutive function Ψ. The second term contains $\hat{\sigma }$, which has a constitutive equation per se, subject to the positive valuedness and the dependence of variables chosen for all of the constitutive equations.

This scheme is likely to allow some improvements of the modeling of damage. In this sense, appropriate generalizations can account for more involved effects through the entropy production ${\hat{\sigma }}_{\mathrm{dam}}$. Indeed, the use of ${\hat{\sigma }}_{\mathrm{dam}}$ looks more flexible than the recourse to the dissipation potential. For instance, in [29] the damage rate, here, $\dot{\varphi }$, is related to a dissipation potential *F* by $\dot{\varphi }=-\partial_{Y}F$, where *Y* is an appropriate density release rate. Hence, terms such as those in (Equation 24) and (Equation 26) account for the rate as (nine) of [29] for the accumulated plastic strain.

Future developments might investigate the use of degradation functions induced by the damage variable so as to describe, e.g., possible anisotropic effects.

## Data Availability

The study did not report any data.

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
