# Peer review of "A Phase-Field Approach to Continuum Damage Mechanics"

_materials, 2022, doi:10.3390/ma15217671_

Round 1
Reviewer 1 Report
All the comments are given in the attached document.

Reviewer 2 Report
Please define all the mathematical characters at its first stage of usage or include the summery of notations at the end of paper.
Reviewer 3 Report
Section 1:
It would benefit the reader if some introduction and context is given here for instance a reference to (R1) Continuum Mechanics book by Gurtin and (R2) The large strain Finite Element Method a Practical Course, Wiley 2015. In addition it should be noted that one is focusing on relatively small strains as indicated by the choice of Green-St. Venant strain measure. Also, one should be explicit to as what stress tensor matrix is used, say Cauchy stress tensor. In addition, it seems that one is dealing with isotropic problems only, see Large strain finite element method a practical course book.
Section 2:
The damage variable is a scalar indicating isotropic damage with all its limitation; damage is usually anisotropic see (R3) “A novel framework for elastoplastic behaviour of anisotropic solids” by Z Lei, CR Bradley … Computational Particle Mechanics 7 (5), 823-838, 2020.
Equation 1, please explain each of the variables.
Equation 2, the same problem as equation 1.
Section 3: Unnumbered equation before the equation 5; now T has become second Piola-Kirchhoff stress matrix, thus invalidating equation 1?
In the light of this, I would suggest that at this point we go back to the physics and avoid unnecessary mathematical jargon detached from physics itself; Equilibrium equations are always written in terms of Cauchy stress matrix, usually termed T (see R1 and R2)
Section 3, 4 and 5: The authors quickly jump from involved mathematical derivations to conclusions. Do we have any physical demonstration of the model? Is there any limitations to the model, its applicability, etc.
May I suggest that the authors go back to structuring the paper in such a way that there is a clear connection to recent developments in the state of the art approaches to material modelling, clearly differentiating between EOS and shear part, between plastic flow and micromechanical damage, see R3. That way the work will be placed in its proper material context with clear implications to its applicability.
I hope that this is of help.
Best regards
Round 2
Reviewer 3 Report
Dear authors,
Thanks for considering my comments; your effort makes the paper acceptable. However, I would still suggest either extending conclusions or adding some examples; otherwise the paper loses its engineering/physics context; this would improve the paper, but I leave the decision to you.
Best regards
